# Targeting Multiple Myeloma through the Biology of Long-Lived Plasma Cells

**DOI:** 10.3390/cancers12082117

**Published:** 2020-07-30

**Authors:** Adam Utley, Brittany Lipchick, Kelvin P. Lee, Mikhail A. Nikiforov

**Affiliations:** 1Department of Cancer Biology, Wake Forest School of Medicine, Winston-Salem, NC 27157, USA; brittany.lipchick@gmail.com; 2Department of Immunology, Roswell Park Comprehensive Cancer Center, Buffalo, NY 14203, USA; kelvin.lee@roswellpark.org

**Keywords:** multiple myeloma (MM), long-lived plasma cell (LLPC), bone marrow microenvironment (BMME)

## Abstract

Multiple myeloma (MM) is a hematological malignancy of terminally differentiated bone marrow (BM) resident B lymphocytes known as plasma cells (PC). PC that reside in the bone marrow include a distinct population of long-lived plasma cells (LLPC) that have the capacity to live for very long periods of time (decades in the human population). LLPC biology is critical for understanding MM disease induction and progression because MM shares many of the same extrinsic and intrinsic survival programs as LLPC. Extrinsic survival signals required for LLPC survival include soluble factors and cellular partners in the bone marrow microenvironment. Intrinsic programs that enhance cellular fidelity are also required for LLPC survival including increased autophagy, metabolic fitness, the unfolded protein response (UPR), and enhanced responsiveness to endoplasmic reticulum (ER) stress. Targeting LLPC cell survival mechanisms have led to standard of care treatments for MM including proteasome inhibition (Bortezomib), steroids (Dexamethasone), and immunomodulatory drugs (Lenalidomide). MM patients that relapse often do so by circumventing LLPC survival pathways targeted by treatment. Understanding the mechanisms by which LLPC are able to survive can allow us insight into the treatment of MM, which allows for the enhancement of therapeutic strategies in MM both at diagnosis and upon patient relapse.

## 1. Introduction

Multiple myeloma (MM) is a bone marrow resident hematological malignancy of antibody-secreting plasma cells. It is the second most common hematological malignancy with 30,000 newly diagnosed cases each year [1]. Although new treatments have shown promising results, myeloma remains incurable. MM is unique among hematological malignancies as it is critically dependent upon the bone marrow microenvironment (BMME) for survival. The foundation for this dependency is in the ability of the bone marrow microenvironment to facilitate survival of the precursor cell subset that develop into MM and long-lived plasma cells (LLPC). LLPC are terminally differentiated antibody secreting B lymphocytes that reside in specialized survival niches in the BMME [2]. LLPC are long-lived as a result of extrinsic pro-survival signals derived from the BMME that go on to regulate cell intrinsic programs of LLPC survival including increased metabolic fitness, the unfolded protein response, and enhanced autophagy [3,4,5,6]. Therefore, understanding the extrinsic and intrinsic survival programs of survival in LLPC that allow them to live for such long periods of time is critical in understanding how anti-MM therapies are effective and how we can augment therapeutic strategies for greater efficacy.

The ability of LLPC to survive for years requires that they successfully compete for space and nutrients in the bone marrow (BM), which is the primary site of hematopoiesis wherein nutrient availability can change dramatically over time. The plasma cell field has demonstrated a critical role for the BM microenvironment in regulating LLPC survival in this dynamic and competitive niche. Extrinsic signals emanating from cellular interactions and soluble factors produced in the BM microenvironment have the ability to regulate survival. However, critical intrinsic survival programs are required for LLPC survival including metabolic fitness and increased autophagy. These cell intrinsic programs facilitate successful competition for survival during the time of nutrient starvation and ensure fidelity under cell stress. The same biology of competition exists for MM, as MM cells exist in the same BMME but also have the necessity to proliferate. Therefore, understanding the mechanisms by which LLPC survive in the BMME will allow insight into how current therapeutic strategies in MM are effective, and give us insight into how we can augment those treatments for increased patient survival.

## 2. B Cell Biology and the Creation of a Plasma Cell

The humoral immune response is built upon the ability of a B cell to become a terminally differentiated antibody secreting a cell known as a plasma cell (PC) [7,8,9]. Bone marrow resident immature B cells undergo a round of negative selection in order to mature [10,11,12]. Mature follicular and marginal zone B cells reside in secondary lymphoid organs and can respond to the antigen in a T cell independent and dependent manner [13,14]. After activation through the B cell receptor (BCR), B cells migrate to newly-formed germinal centers where they undergo class switch recombination and somatic hypermutation and can differentiate into memory B cells or antibody secreting plasma cells [15,16,17]. It is at this point that BCR signal strength and the quality/quantity of “danger” signals (toll like receptor (TLR) agonists, or other pathogen associated signals through nod-like/rig-like receptors (NLRs, RLRs)) [18,19] can program distinct functional outcomes in the quality and persistence of plasma cells [20,21].

Several cell intrinsic programs are induced during the transition from an activated B cell to a plasma cell. Upon B cell activation, a metabolic shift occurs in which the cells increase glucose uptake and induce mitochondrial respiration, which increases both the tricarboxylic acid cycle (TCA) and oxidative phosphorylation [22]. This metabolic program facilitates the acquisition of an effector function in cytokine production and immunoglobulin synthesis. A byproduct of mitochondrial respiration is the production of reactive oxygen species (ROS). ROS are well characterized in their capacity to damage cells [23,24]. In order to maintain proper levels of ROS, many cells upregulate antioxidant enzymes such as glutathione peroxidase (GPX), superoxide dismutase (SOD), and catalase (CAT) that act to scavenge ROS from the cellular system [25]. However, mitochondrial respiration-derived ROS are required for B cells to differentiate into plasma cells [26]. This suggests a possible role for ROS in cell signaling. ROS have been shown to inactivate the phosphatase PP2A, the negative regulator of Akt [27], and directly activate the transcription factor NF-κB [28] in which both have been shown to be critical for B cell development [29,30]. Upon B cell differentiation into plasma cells, distinct functional capacity is provided by different isotypes of antibodies secreted by the plasma cells [31].

IgM secreting plasma cells are the first to produce an antibody with the fastest response and a lower affinity binding capacity [32]. IgE secreting plasma cells are most generally associated with allergic reactions, but also function in the clearance of parasitic worms [33,34]. IgA is generally localized to the mucosal tissues and maintain homeostasis of the gut microbiota [35]. IgG production facilitates the longest duration of protection due to the high affinity for antigen and stability of the IgG molecules in circulation [36]. IgG production is canonically associated with long-lived plasma cells (LLPC) with IgM associated with short-lived plasma cells (SLPC).

B cells can differentiate into SLPC that reside primarily in secondary lymphoid organs such as the spleen and primarily secrete IgM [37]. Memory B cells can also differentiate into SLPC with persistent antigen exposure [38]. However, humoral immunity can be sustained in the absence of memory B cells [37,39] or continual antigen availability [40,41]. The half-life of circulating antibody molecules is approximately days to weeks [42] while the half-life of antibody titers may last years to decades in the human population [43]. Therefore, a second non-mutually exclusive model has been proposed wherein B cells differentiate into mostly IgG secreting LLPC, which are home to the bone marrow and can live for the lifetime of an organism (Figure 1) [37,44]. Although a specific phenotype that distinguishes SLPC from LLPC/MM cells has remained elusive, LLPC have increased expression of the transcription factors BLIMP1, XBP1, and IRF4 [45], but specific transcription factors necessary for PC development in certain inflammatory contexts are still being discovered as was the case for T-bet in a recent publication [46]. It has also been suggested that initial transcriptional programming through Zbtb20 may be critical in the induction of an LLPC [47]. However, whether levels of Zbtb20 are stable upon LLPC differentiation or must be maintained by other signals in the bone marrow microenvironment is still unclear. High cell surface expression of CD93 has also been demonstrated to distinguish LLPC from SLPC and play a role in maintaining bone marrow plasma cells (PC) [48]. It is also clear that the induction of PC from B cells is coupled with specific epigenetic programming [49,50]. However, how these intrinsic transcriptional and epigenetic programs are initiated and maintained in the LLPC pool that resides in the complex bone marrow microenvironment is not fully understood.

## 3. Long-Lived Plasma Cell Induction and Maintenance

Understanding the mechanisms by which LLPC are induced and are capable of long-term survival is important in both our ability to design effective vaccines as well as alleviate antibody-mediated autoimmune disease. In seminal experiments, when tracking the half-life of protective circulating antibody titers in the human population after vaccination, it was determined that the half-life of measles titers is on the order of 3000 years [43]. During B cell differentiation, a PC must be able to “unlock” the ability to respond to survival signals in the bone marrow microenvironment. This is represented by the induced transcriptional network and epigenetic remodeling during B cell to plasma cell differentiation that facilitate the upregulation of factors necessary for plasma cell function and survival [49,51]. Furthermore, the plasma cell must be able to move to the bone marrow survival niches to access those signals. Understanding how a plasma cell that can move to the bone marrow and respond to signals becomes an LLPC is, therefore, critical in our ability to elicit an effective durable humoral immune response and fill a major gap in the field. This gap answers how a PC becomes long-lived.

LLPCs are not intrinsically long-lived. Rather, they depend upon extrinsic signals from specialized niches in the bone marrow for their continued persistence [44]. The LLPC survival niches are found predominantly in the bone marrow (BM), despite the fact that other anatomical locations have been described [52,53,54]. Both cellular and soluble extrinsic components of the LLPC niche have been described as well as necessary intrinsic mechanisms by which LLPC survival is maintained in the BM microenvironment. Many of these extrinsic and intrinsic mechanisms of LLPC survival are shared with MM. Therefore, an understanding of these survival signals and pathways is critical for our understanding of both LLPC biology as well as that of MM and lends translational relevance to the fundamental understanding of the mechanisms of LLPC survival.

### 3.1. Cellular Partners

Cellular and stromal partners in the niche support LLPC survival by providing an infrastructure for stable maintenance in the BM microenvironment through chemokine production as well as direct survival cues via cytokine production. The primary chemokine regulating PC homing to the BM niche is CXCR4, which is where PC becomes responsive during differentiation [55,56]. Although the mechanisms differ by which each cellular partner facilitate LLPC survival, the common thread in the literature is in understanding that, in the absence of specific cellular components of the niche that provide support for LLPC survival, LLPC cannot be maintained. For LLPC to successfully survive, they require each component of the niche to be intact.

CXCL12-Abundant Reticular Cells (CAR cells) facilitate the homing of LLPC to the survival niche and provide a platform in the bone marrow niche for LLPC survival [57]. Eosinophils have the capacity to produce cytokines known to regulate PC survival including APRIL (a proliferation-inducing ligand) and IL-6 [58]. Though direct interaction was not demonstrated to be required for LLPC survival, eosinophils are thought to occupy space within the LLPC survival niche. Mesenchymal stem cells (MSC) are also thought to provide APRIL in the bone marrow microenvironment to facilitate LLPC survival [59]. MSC have also been demonstrated to produce IL-6 in a cell-contact dependent manner when cultured in vitro with PC derived from the bone marrow, which suggests an important axis by which extrinsic interactions between PC and MSC can regulate cell-intrinsic PC survival (through IL-6 signaling-mediated transcriptional regulation of PC survival) [60]. In support of a critical role for MSC-derived IL-6 in long-lived plasma cell survival, it has been shown that IL-6 can facilitate the in vitro generation of human long-lived plasma cells in conjunction with other stromal-cell-derived soluble factors [61]. 

Recent work has demonstrated a role for T regulatory cells in maintaining the plasma cell niche [62]. T regulatory cells were required for LLPC survival. However, CTLA4, which is a negative costimulatory molecule, was shown to negatively regulate the size of the LLPC niche since its conditional deletion from the T regulatory cell subset caused an increase in the number and percentage of PC in the bone marrow. Since CTLA4 is known to interact with and induce signaling downstream of CD80 and CD86 on dendritic cells to induce IL-6 and indoleamine 2,3 dioxygenase (IDO) [63], which is an immunosuppressive enzyme, it is possible that a dynamic interplay between each of these three cell subsets regulates the balance between both survival and available space. Megakaryocytes also provide survival signals to LLPC as well as dendritic cells themselves [58,62,64,65,66,67]. Though most of these cells may act by producing soluble factors, which facilitate LLPC survival including APRIL, BAFF, and IL-6. The cellular and molecular interactions involved in extrinsic signaling into intrinsic programs of survival are less well known (Figure 2) [60,68].

### 3.2. Cell Intrinsic Programs

Along with the extrinsic survival components of the LLPC niche are cell intrinsic programs that specifically support the survival of LLPC. As highly biosynthetic cells, LLPC must be able to both handle a high degree of protein turnover as well as have plenty of resources to produce the protein. The unfolded protein response is a cell intrinsic program necessary for B cells to differentiate into a PC and allows a PC to handle the cell stress induced by the differentiation process through the spliced form of the protein XBP-1, but it is equally necessary for B cells to differentiate into SLPC and LLPC [69,70,71]. Two cell intrinsic programs that facilitate the ability of LLPC to meet the requirements for biosynthesis/fidelity under nutrient availability and distinguish LLPC from SLPC are autophagy and metabolic fitness.

#### 3.2.1. Autophagy

Autophagy is a highly conserved process by which cells under stress can induce “self-eating” in order to survive in the absence of nutrient availability or dispose of misfolded protein cargo. Autophagy was first described as a process by which mitochondria could be turned over during times of nutrient stress in rat liver cells [72]. The ability of autophagy to regulate the turnover of dysfunctional mitochondria or provide the infrastructure for mitochondrial maintenance is a potential mechanism by which autophagy itself can regulate metabolism when, classically, the model is such that nutrient starvation regulates autophagy. However, autophagy is not relegated to the turnover of mitochondria, but can facilitate the degradation of misfolded proteins, intracellular organelles, endoplasmic reticulum (ER), or clearance of intracellular pathogens [73]. The capacity for autophagy to regulate ER is especially important in PC since a hallmark of PC differentiation is an expanded ER in order to facilitate continual antibody production. Increased basal levels of autophagy distinguish LLPC from SLPC and autophagy is required for LLPC survival and antibody production in the BM niche [5].

The signaling pathways involved in the regulation of autophagy converge on the mTORC1 (mammalian target of rapamycin 1)/AMPK pathways [74]. mTOR was initially discovered by David Sabatini and Michael Hall [75] and has since been described as a molecular rheostat in cell biology, which senses various growth factors and nutrient availability in order to regulate various cell processes including growth, division, metabolism, and immune cell differentiation [76]. Inhibition of mTOR by rapamycin has been shown to sensitize myeloma cells to Dexamethasone, which is one of a few frontline therapies used in the clinic [77]. The negative regulator of mTOR is AMPK, which can directly sense AMP/ATP levels in the cell and can be directly activated through calcium-mediated signaling [78,79], and, therefore, the mTOR/AMPK axis can integrate a diverse set of stimuli in regulating the induction of autophagy [80]. This signaling axis has been demonstrated to be active and targetable in MM [81] and may lead to new therapeutic strategies in combination with proteasome inhibitors and immunomodulatory drugs. The signaling pathways that regulate mTOR and AMPK for the induction of autophagy have been well described in the T cell literature as downstream of the T cell receptor and CD28 signaling, which is the prototypic T cell costimulatory molecule [82]. Through both direct signaling from Ca++ downstream of PLCγ and through nutrient sensing, AMPK can be activated and can facilitate the induction of autophagy. Although autophagy itself has been described as critical for LLPC survival and antibody production, how it is regulated in LLPC, whether directly through signaling or nutrient availability, is unknown. Since we do not know how autophagy is regulated by or possibly important in the regulation of LLPC metabolism, it is important to understand that both of these cell intrinsic pathways for LLPC survival may crosstalk between one another and how we can then use this knowledge in targeting MM.

#### 3.2.2. Metabolic Fitness

The highly biosynthetic nature of LLPC requires both high nutrient availability and energy production for continual antibody synthesis. Antibody molecules are also heavily glycosylated, which require glucose availability in order to make the necessary post-translational modifications for antibody secretion [3]. LLPC reside in the BM, which is the primary site of hematopoiesis, where nutrient availability may be limited at times and, therefore, create acute cell stress. It is, therefore, critical that LLPC have a high degree of metabolic fitness in order to survive and produce the antibody. Recent work has demonstrated that high levels of glucose uptake is a defining feature of LLPC that distinguish them from SLPC [4]. The metabolic programs that utilize glucose are critical to our understanding of LLPC metabolic fitness.

Glycolysis is the metabolic process by which glucose, a 6-carbon sugar molecule, is broken down into two 2-carbon molecules of pyruvate [83]. Although complex in the biochemistry facilitating this breakdown, it is more the fate of the glucose-derived products that are important in regulating plasma cell nutrient availability and energy production. Pyruvate has one of two well characterized fates. In many tumor cells in which glycolysis is induced independently of oxygen availability, pyruvate is often converted to lactate and shunted out of the cell in a process of aerobic glycolysis known as the Warburg Effect [84]. However, this is less efficient for energy production with a net gain of only two ATP than if the pyruvate is used in the second major pathway through mitochondrial respiration for a net gain of 36 ATP [83]. 

In LLPC, pyruvate is able to be metabolized through this mitochondrial pathway. Pyruvate is imported through the mitochondrial pyruvate importer protein in LLPC and used for energy production when LLPC are under stress conditions [3]. When the mitochondrial pyruvate importer is genetically knocked out after vaccination, there is a progressive loss of antigen-specific antibody titers and LLPC numbers in the BM, which suggests that glucose-derived pyruvate is required for LLPC survival over times of nutrient stress in the BM and that the ability to access this metabolic program during times of cell stress is what distinguished LLPC metabolic fitness from SLPC [3]. What is unclear is how this metabolic fitness is regulated in LLPC. MM requires the same degree of metabolic fitness as LLPC with the additional nutrient requirements of cell division. Although MM has high expression levels of glycolytic enzymes and engages in glycolysis [85,86], it is becoming more evident that mitochondrial metabolism plays a major role in myeloma [87]. A greater understanding of metabolic regulation in LLPC may provide new insight into metabolic targets in MM.

#### 3.2.3. The Intersection of LLPC and MM

As seen in previous sections, many of the treatments that have shown clinical efficacy in multiple myeloma target the intrinsic programs of survival shared with LLPC or the cellular partners in the BM microenvironment. Due to the highly biosynthetic nature of LLPC/myeloma, drugs that target protein degradation through proteasome inhibition have been highly effective in MM treatment even though it was initially thought that the mechanism by which proteasome inhibitors killed MM was through the inhibition of the transcription factor NF-κB [88]. The importance of cellular components of the BM microenvironment for multiple myeloma cell survival is demonstrated in the efficacy of immunomodulatory drugs including Dexamethazone (a corticosteroid) and the immunomodulatory drugs Lenolidamide and Pomalidomide, which both target cereblon, an E3 ubiquitin ligase that regulates T cell function [89].

Multiple myeloma is critically dependent on the bone marrow microenvironment and shares many soluble mediators of survival with LLPC (Figure 2). IL-6 has been shown to both augment myeloma cell growth and provide a survival signal, which inhibits apoptosis [90]. BCMA is the receptor for APRIL/BAFF, known survival factors for LLPC, which is overexpressed on myeloma cells [91]. BAFF can also signal specifically through the BAFF receptor (BAFF-R), and the shared receptor with APRIL in TACI [92]. The ability of BAFF to regulate B cell and PC survival through different receptors may be partly dependent on the temporal regulation of receptor expression [93]. By activating BAFF, BCMA activates several survival and growth pathways in multiple myeloma including NF-κB, Akt, and MAPK signaling [94]. Cellular partners like stromal cells are able to regulate myeloma survival by producing cytokines like APRIL and BAFF as well as delivering exosomes that inhibit pathways known to regulate apoptosis (JNK signaling) [95,96]. Integrin signaling can also drive myeloma cell survival, adhesion, and invasion in the bone marrow microenvironment, which are all hallmarks of multiple myeloma disease progression [97,98]. Certain anti-apoptotic proteins have also been described in myeloma cell survival, namely Mcl-1, which has also been shown to be the primary Bcl-2 family member responsible for LLPC survival [99,100].

Targeting MM survival pathways through receptors like BCMA has demonstrated clinical success with several emerging strategies. Chimeric antigen receptor T cells (CAR-T cells) are T cells that are expanded with a genetic construct that facilitates the expression of artificial receptors, which recognize specific antigens. Clinical trials with anti-BCMA CAR-T cells have shown efficacy in certain patients [101]. Other potential targets for CAR-T cell therapy include CD138 [102], CD38 [103], and SLAMF7 [104], which are all cell surface proteins expressed on MM cells. Other strategies include monoclonal antibodies directed to cell surface receptors with current drugs approved for targeting SLAMF7 [105] and BCMA [106]. It is also possible that treatments, which target MM could be used in autoimmune disorders driven by plasma cell production of auto-antibodies.

#### 3.2.4. CD28: Bridging the BMME and Intrinsic Survival Programs in LLPC/MM

During B cell differentiation, a specific transcriptional network is activated including genes necessary for plasma cell survival and function. The lineage defining the transcriptional regulator of B cells is PAX5, which actively represses expression of *Prdm1*, the gene encoding BLIMP1. This is the master regulator of plasma cell identity [107,108]. BLIMP1 is required for antibody secretion and regulates the unfolded protein response (UPR) [109]. Many other genes necessary for PC survival and function are upregulated during B cell differentiation including *Irf4, Xpb1* (the regulator of the UPR)*,* and *Cd28* [51,109,110]. CD28 is the canonical T cell costimulatory receptor [111,112]. 

In conjunction with T cell receptor (TCR) activation, CD28 co-stimulation through engagement with its cognate ligands CD80/CD86 on antigen presenting cells (APC) augments proliferation, cytokine production, and survival during the transition to effector T cells [113,114,115,116,117]. CD28 is also expressed on the malignant BM-resident PC in multiple myeloma (MM) [118,119] and normal PC [120], but its function in B lineage has not been well characterized. We have previously shown in MM that CD28 activation by itself transduces a major pro-survival/chemotherapy resistance signal [121,122], and others have shown that CD28 signaling in MM can decrease MM cell susceptibility to CD8 T cell-mediated anti-tumor immune responses [123]. However, its function in normal PC is largely uncharacterized. Genetic knockdown or pharmacological inhibition of CD28 has been shown to decrease humoral responses to many pathogenic challenges [124,125,126,127,128,129,130,131,132,133], which suggests that CD28 plays a prominent regulatory role in plasma cell biology. Therefore, understanding the mechanism by which CD28 activation by the extrinsic bone marrow microenvironment is able to drive a cell intrinsic program of LLPC/MM survival would advance the field by allowing us to understand the extrinsic interactions in the BM that govern cell intrinsic programs of survival in order to augment vaccine design, alleviate autoimmunity, and treat MM.

Activated T cells require increased metabolism to meet their biosynthetic needs for effector functionality and survival [134,135,136]. This includes the CD28-mediated increase in glucose uptake by upregulating the glucose transporter GLUT1 [137]. CD28 has also been shown to regulate the induction of glycolysis for cell growth and proliferation and the upregulation of mitochondrial respiration for long-term survival [137,138]. CD28 regulates the longevity of memory T cells through reorganization of mitochondrial morphology and enhanced mitochondrial spare respiratory capacity, which is a hallmark of memory T cell metabolism [139]. Mitochondrial respiration is required for T cell activation, proliferation, and differentiation through reactive oxygen species (ROS)-dependent signaling [140]. CD28-mediated ROS signaling in T cells is also necessary for NF-κB dependent IL-2 production [141]. The transcription factor IRF4 is a target of NF-κB and is upregulated during B cell to PC differentiation, and is required for plasma cell survival [109,142]. IRF4 also regulates metabolic programming in T cells by specifically regulating glucose uptake, mitochondrial mass, and mitochondrial respiration [143,144], which suggests that it may be downstream of CD28 activation in the T cell context. Since CD28 has the capacity to govern essential components of the LLPC program, it makes a good target for interrogation in both LLPC and MM biology.

We have previously reported that CD28 is expressed on plasma cells and that its activation through an interaction with CD80/86 expressing DC in the bone marrow microenvironment is required for bone marrow-resident LLPC survival in vitro and in vivo but has no effect on SLPC survival [145]. In our studies, we use anatomical location to equivocate bone marrow plasma cells to the long-lived plasma cell subset, and splenic plasma cells as the short-lived compartment with the caveat that both compartments are heterogeneous. Two binding motifs have been described on the CD28 cytoplasmic tail that regulate several distinct signaling pathways and are phosphorylated upon receptor activation to illicit distinct functional outcomes [124,126,146]. Phosphorylation of the membrane proximal Y^170^MNM motif induces binding of the SH2 domain of the p85 subunit of phosphatidyl-inositol 3-kinase (PI3K) and activation of the downstream PI3K → PDK1 → Akt → NF-κB signaling pathway [146]. Phosphorylation of the C-terminal P^187^YAP^190^ proline motif leads to Lck recruitment, and then the SH3-mediated recruitment of Grb2/Vav, which leads to Rac1/Cdc42 → ras → AP-1 and PLCγ → NF-κB /NFAT pathways [146]. The downstream pathways from CD28 that govern LLPC survival had not been described. However, because CD28 is specifically capable of inducing a pro-survival signal in LLPC, but not SLPC, discovering the specific downstream mediators of CD28 survival signaling will allow us to define the molecular mechanisms by which LLPC can respond to CD28 engagement, and, therefore, how a PC becomes long-lived. By establishing this axis of survival signaling downstream of CD28, we may also be able to target this axis to augment the treatment of MM.

We have recently published that CD28 pro-survival signaling through Grb2/Vav/SLP-76/NF-κB regulates LLPC metabolic fitness by increasing the expression of the transcription factor IRF4 [147]. By increasing glucose uptake and mitochondrial respiration, CD28 is able to induce a metabolic program in LLPC that allows them to survive in the bone marrow microenvironment under conditions where nutrient availability may be limiting. The CD28-mediated increase in mitochondrial respiration drives higher levels of reactive oxygen species, which are themselves critical in relaying the CD28 pro-survival signal. In this way, the byproducts of CD28-mediated metabolic regulation feed the survival signal emanating from the extrinsic bone marrow microenvironment (Figure 3).

CD28 is an example of how extrinsic signals in the bone marrow microenvironment are able to drive cell intrinsic programs of survival (at the transcriptional and metabolic levels). Much of the transcriptional network that mediates LLPC survival is driven by extrinsic signaling (CD28-IRF4-metabolic fitness, BAFF-Mcl1, IL6-STAT3). In LLPC biology, the bone marrow microenvironment is the critical component in driving the transcriptional and physiological processes that facilitate LLPC persistence.

CD86, which is the cognate ligand for the prototypic T cell co-stimulatory molecule CD28, is expressed by myeloma cells and is required for their survival [148]. Since we have shown that CD28 itself is important in myeloma progression and survival [121,122,149], this CD28/CD86 axis may represent a possible mechanism by which cis-interactions in the myeloma cell microenvironment may facilitate survival and disease progression.

## 4. Conclusions

Since multiple myeloma is the malignancy of LLPC, and LLPC can live for the lifetime of an organism, myeloma cells can access an immortality program before any genetic mutations are present. Since multiple myeloma shares the same bone marrow niche requirements as LLPC, myeloma cells must compete for nutrients in the same environment to the same extent as LLPC and similarly respond to survival signals. Multiple myeloma cells also continue to secrete antibodies, which requires high levels of nutrient uptake, autophagy, and responses to ER stress. This makes the survival demands of MM in the BM microenvironment even more substantial than that of LLPC. Therefore, finding the mechanisms by which myeloma cells can maintain the survival programs built into LLPC through interactions with the extrinsic survival signals in the BM and cell intrinsic programs would allow us to target those interactions for disruption of multiple myeloma survival and, therefore, augment the current treatments available for myeloma patients.

## Figures and Tables

**Figure 1 cancers-12-02117-f001:**
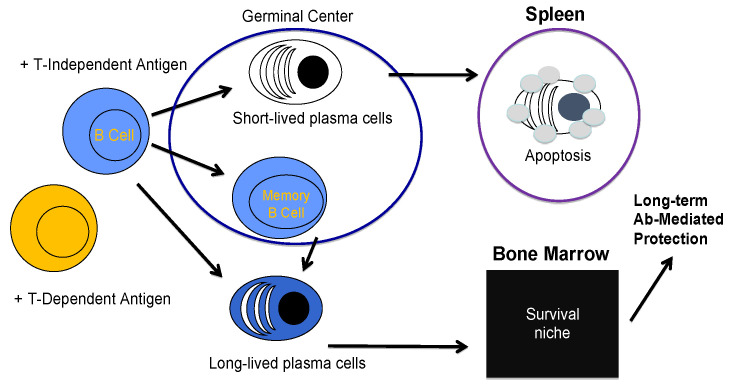
Model of B to plasma cell (PC) differentiation. Upon activation with antigen in a T cell independent (above) or dependent (below) manner, B cells can differentiate into antibody-secreting plasma cells that either reside in secondary lymphoid organs such as the spleen, where they primarily secrete IgM and live for days to weeks before dying by apoptosis. B cells can also differentiate into memory B cells, which, upon reactivation with the antigen, can differentiate into plasma cells. In a second non-mutually exclusive model, B cells can differentiate into long-lived plasma cells that are home to the bone marrow where they occupy special survival niches and provide durable humoral immunity through the long-term production of antigen specific antibodies, primarily IgG.

**Figure 2 cancers-12-02117-f002:**
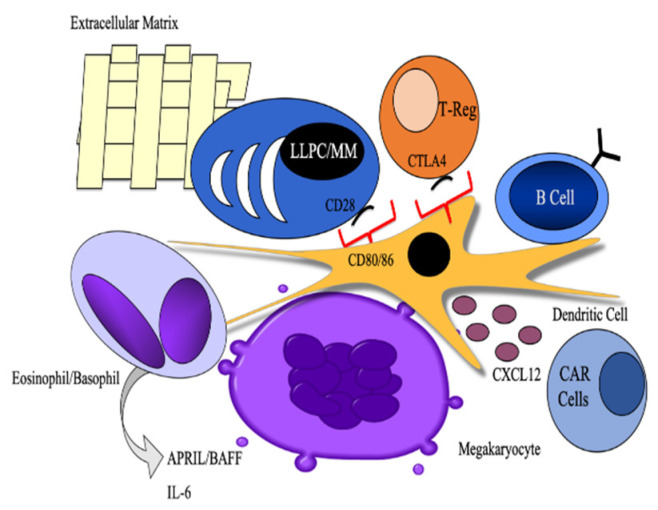
Model the cellular partners and soluble survival signals in the bone marrow microenvironment shared by both (long-lived plasma cells (LLPC) and multiple myeloma (MM). Cellular partners like that of eosinophils, basophils, dendritic xells (pictured in center), CXCL12-abundant reticular (CAR) cells, T regulatory cells, and B cells can provide important cell-cell interactions for the survival of LLPC along with factors held in the extracellular matrix (ECM). Survival signals emanating from these accessory cell subsets include IL-6, APRIL, BAFF, and homing signals like CXCL12. Interactions between CD28 on LLPC/MM and CD80/CD86 on DC and other myeloid cells can provide survival signals to both LLPC and MM, and promote dendritic cells (DC) production of IL-6 and IDO.

**Figure 3 cancers-12-02117-f003:**
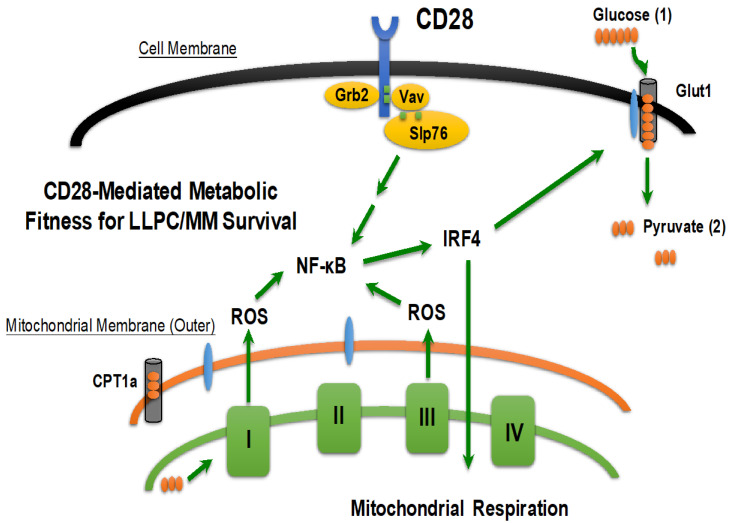
Model of CD28-mediated metabolic fitness in LLPC. CD28, through Grb2/Vav/SLP-76 signaling in LLPC, induces NF-κB activation. NF-κB is a known transcriptional regulator of IRF4, which is a transcription factor required for LLPC survival. CD28 activation increases IRF4 expression, which increases LLPC glucose uptake, mitochondrial mass, and mitochondrial respiration. One byproduct of mitochondrial respiration is the production of reactive oxygen species, which go on to facilitate further NF-κB activation for IRF4-mediated metabolic fitness and LLPC survival.

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
