# Peer review of "Targeting Multiple Myeloma through the Biology of Long-Lived Plasma Cells"

_cancers, 2020, doi:10.3390/cancers12082117_

Round 1
Reviewer 1 Report
I am satisfied that the authors have modified the manuscript accordingly
Reviewer 2 Report
The authors have clarified several of the questions I raised in my previous review. Most of the major problems have been addressed by this revision.
No further comment from this reviewer.
Reviewer 3 Report
accept
This manuscript is a resubmission of an earlier submission. The following is a list of the peer review reports and author responses from that submission.
Round 1
Reviewer 1 Report
This is an interesting review of the biology of long lived plasma cells and how understanding this biology might be linked to the treatment of multiple myeloma. The paper is well written.
Comments
Specify the LLPC phenotype, and the difference between, SLPC, and Myeloma cells. ( CD93?)
Add a chapter on epigenetic regulation and transcriptional factors (STAT3)
At line 57 the authors state that “Immature B cells can be activated by T cell independent type 1 antigens (known as transitional type 1, T1 cells) such as lipopolysaccharide (LPS) or type 2 antigens, known as type 2 transitional B cells (T2), but generally undergo a round of negative selection in order to mature [10, 11].” It is not clear if the authors are talking about T cell independent types or about immature B-cells and negative selection in the BM or about transitional B-cell subsets? Please, rephrase this paragraph in order to be more definite/informative.
At line 65 the authors state that “It is at this point that BCR signal strength and the quality/quantity of “danger” signals can program distinct functional outcomes in the quality and persistence of plasma cells that are created [17, 18].” It is not clear what authors mean by “danger” signals, may be explain by giving examples? As these signals (from BCR and other receptors) are important for the quality and persistence of plasma cells, the authors could give a little bit more details on the implicated mechanisms.
At line 113 the authors state that “LLPCs are not intrinsically long-lived; rather, they depend upon extrinsic signals from specialized niches in the bone marrow for their continued persistence [38].”. B-cell intrinsic factors were also suggested to play a role including a preprogramation of long survival mediated by specific transcription factors such as ZBTB20 (PMID: 24711583) and T-Bet (PMID: 31076359).
P 122 3.1 Cellular partners
we need more details on MSCs (secreted factors) and the role of dendritic cells
At line 235 there is a typo error: “Many of the same extrinsic and extrinsic programs of LLPC survival govern the survival of the…”. Many of the same intrinsic and extrinsic programs?
p254 specify the difference between BAFF and April on BCMA, and the BCMA /MCL1 relationship in survival
In the section “The intersection of LLPC and MM”, the authors should discuss the recent advances in MM therapy targeting the receptors involved in LLPC survival such as BCMA CAR-T cells and monoclonal antibodies.
Reviewer 2 Report
The authors present a comprehensive review of long lived plasma cell biology and how this knowledge can inform potential therapeutic avenues for Multiple Myeloma. The review is well organised and presented.
Minor comment:
Figure legends could be more explanatory particularly for Figure 1
Reviewer 3 Report
The review provides an interesting and detailed overview of long-lived plasma cell (LLPC) biology and of critical extrinsic and intrinsic survival programs by which LLPCs survive in the bone marrow microenvironment. Understanding these mechanisms could improve our knowledge about MM bone marrow niche and how anti-MM therapies are effective and how we can improve therapeutic management for greater efficacy.
I have only two minor comments:
- At page 6, line 235, the word extrinsic is repeated twice.
- At page 8, line 324, the authors have reported some important papers which demonstrate the important role of CD28 in MM progression and survival. I think that an interesting paper Leone et al. Blood 2015 (PMID: 26185130) has been overlooked. It adds a new piece of information to the role that CD28 plays in plasma cell biology, demonstrating that the binding of CD28 expressed on tumor plasma cells with its ligands CD80/CD86 expressed on bone marrow myeloid dendritic cells protects tumor plasma cells from CD8+ T-cell killing and promotes myeloma disease progression. I feel this paper should be quoted in the review.
Reviewer 4 Report
The authors review the knowledge on the long live plasma cells (LLPC) in the bone marrow. This is an important topic indeed because all myeloma patient eventually relapses and it may be related to the so called « myeloma stem cells » that have not been eradicated.
They first describe the normal physiology of B cell differentiation and then focus on normal long live plasma cell formation. They report the extrinsic and intrinsic signals as well as the cellular partners who have a role in the biology of LLPC. There is a long paragraph on the role of CD28.
Major revisions
It is not clear what are these LLPC ? « myeloma stem cells » ? How are these cells charactarized ? specific phenotype ? what is the link between the myeloma clonal plasma cells and these LLPC ?
Add 3 figures :
1/one for what is happening in the myeloma bone marrow microenvironment. Show the interactions between APRIL/BAFF and BCMA on myeloma cells and so on…
2/ a second figure on CD28 would help
3/ a third one on how the current treatments could target these LLPC
From line 236 to 243 move it to introduction
Minor revisions
English should be checked : some sentences begin with « And as… » (line 330). « Because we have shown… », « As recent work has demonstrated.. » and editing : « where nutrient availability may me limited »
Fig 1 : are you sure it is both T independant antigen ? please show the T dependant pathway
Fig 2 : stromal cell or dendritic cell ? it looks like DC are missing in the figure ? what are the yellow blocks in the back ? In the figure legend, explain what CAR is : in the current context, we think of chimeric antigen T cells ! what are the receptors between LLPC/MM and eosinophils ?
Ref 1 : authors are missing
